# A Computational Approach to Investigate TDP-43 RNA-Recognition Motif 2 C-Terminal Fragments Aggregation in Amyotrophic Lateral Sclerosis

**DOI:** 10.3390/biom11121905

**Published:** 2021-12-19

**Authors:** Greta Grassmann, Mattia Miotto, Lorenzo Di Rienzo, Federico Salaris, Beatrice Silvestri, Elsa Zacco, Alessandro Rosa, Gian Gaetano Tartaglia, Giancarlo Ruocco, Edoardo Milanetti

**Affiliations:** 1Department of Physics and Astronomy, University of Bologna, Viale Carlo Berti Pichat 6/2, 40127 Bologna, Italy; greta.grassmann@studio.unibo.it or; 2Center for Life Nano- & Neuro-Science, Istituto Italiano di Tecnologia, Viale Regina Elena 291, 00161 Rome, Italy; mattia.miotto@iit.it (M.M.); lorenzo.dirienzo@iit.it (L.D.R.); federico.salaris@uniroma1.it (F.S.); beatrice.silvestri@uniroma1.it (B.S.); alessandro.rosa@uniroma1.it (A.R.); giangaetano.tartaglia@uniroma1.it (G.G.T.); giancarlo.ruocco@roma1.infn.it (G.R.); 3Department of Biology and Biotechnologies “Charles Darwin”, Sapienza University of Rome, Piazzale Aldo Moro 5, 00185 Rome, Italy; 4Department of Neuroscience and Brain Technologies, Istituto Italiano di Tecnologia, Via Morego 30, 16163 Genoa, Italy; elsa.zacco@iit.it; 5Center for Human Technologies, Via Enrico Melen 83, 16152 Genova, Italy; 6Department of Physics, Sapienza University of Rome, Piazzale Aldo Moro 5, 00185 Rome, Italy

**Keywords:** molecular dynamics simulation, protein aggregation, binding regions, TDP-43, Amyotrophic Lateral Sclerosis

## Abstract

Many of the molecular mechanisms underlying the pathological aggregation of proteins observed in neurodegenerative diseases are still not fully understood. Among the aggregate-associated diseases, Amyotrophic Lateral Sclerosis (ALS) is of relevant importance. In fact, although understanding the processes that cause the disease is still an open challenge, its relationship with protein aggregation is widely known. In particular, human TDP-43, an RNA/DNA binding protein, is a major component of the pathological cytoplasmic inclusions observed in ALS patients. Indeed, the deposition of the phosphorylated full-length TDP-43 in spinal cord cells has been widely studied. Moreover, it has also been shown that the brain cortex presents an accumulation of phosphorylated C-terminal fragments (CTFs). Even if it is debated whether the aggregation of CTFs represents a primary cause of ALS, it is a hallmark of TDP-43 related neurodegeneration in the brain. Here, we investigate the CTFs aggregation process, providing a computational model of interaction based on the evaluation of shape complementarity at the molecular interfaces. To this end, extensive Molecular Dynamics (MD) simulations were conducted for different types of protein fragments, with the aim of exploring the equilibrium conformations. Adopting a newly developed approach based on Zernike polynomials, able to find complementary regions in the molecular surface, we sampled a large set of solvent-exposed portions of CTFs structures as obtained from MD simulations. Our analysis proposes and assesses a set of possible association mechanisms between the CTFs, which could drive the aggregation process of the CTFs. To further evaluate the structural details of such associations, we perform molecular docking and additional MD simulations to propose possible complexes and assess their stability, focusing on complexes whose interacting regions are both characterized by a high shape complementarity and involve β3 and β5 strands at their interfaces.

## 1. Introduction

The investigation of the molecular mechanisms that lead to the accumulations of aggregated proteins is crucial for understanding the pathophysiology of many neurodegenerative diseases [1]. Indeed, the accumulation of aggregates containing the DNA- and RNA-binding protein TDP-43 in the central nervous system is a common feature in diseases, such as Amyotrophic Lateral Sclerosis (ALS), Frontotemporal Dementia (FTD), and Alzheimer’s Disease (AD) [2,3]. However, the mechanisms of aggregation are not yet fully understood and various aggregation models have been proposed [4]. In this scenario, the involvement of TDP-43 C-terminal fragments (CTFs) in the molecular mechanisms causing the formation of aggregates has already been widely confirmed [5,6,7,8,9,10].

TDP-43 is composed of an N-terminal domain (NTD), two RNA recognition motifs (RRMs), and a long C-terminal (CTD) glycine-rich region [5], as one can see from Figure 1a. During neurodegenerative diseases, TDP-43 undergoes a wide array of post-translational modifications, including phosphorylation, acetylation, ubiquitination, oxidation, and cleavage [11]. In this study, we are going to focus on two cleavages of the full TDP-43, which give rise to two different CTFs [12]. The CTFs are formed by either residue range 209–414 or 220–414 of TDP-43 [8], corresponding to a portion of the full protein including only the CTD together with a truncated RRM2 fragment (see Figure 1b). Two main categories of CTFs can be usually found, depending on the type of RRM2 constituting them [13]: one is truncated at the residue in position 208 and the other is truncated at residue 219. We call the corresponding truncated RRM2 Fragment A and B, respectively. We note that Fragment A and B comprise two residues that are found to be targeted by post-translational modifications, i.e., cysteine 244 and lysine 263 [11,12].

In normal conditions, the NTD-driven head-to-tail oligomerization spatially separates the high aggregation-prone CTDs of consecutive TDP-43 monomers, antagonizing aggregation [7]. However, if a proteolytic cleavage releases CTFs, these free portions of the protein are able to aggregate [8]: according to current knowledge, the formation of inclusions seems indeed related to this disruption of the physiological oligomerization of TDP-43. Furthermore, the removal of the N-terminus increases the cytoplasmic localization since it deprives the CTF of the Nuclear Localization Signal (NLS) [5].

Although the interaction mechanisms among TDP-43 proteins are still poorly understood, it is presumed that aggregation involves the CTD, which is intrinsically disordered and aggregation-prone, and harbors most of the mutations related to ALS [10]. Thus, it has been discussed how the CTD is necessary for cytoplasmic aggregation and toxicity but not sufficient, as it requires an intact RRM, i.e., the RRM2 fragment in the CTFs is fundamental for this model of aggregation [14]. In physiological conditions, RRM2 is a highly stable domain, due to a cluster of twelve connected hydrophobic residues in its core [9]. However, the cleavage deprives RRM2 of its stabilizing interactions with RRM1. In fact, a study of the RRM2 unfolding after separation from RRM1 found that the mutually stabilizing interaction between RRM1 and RRM2 reduces the population of an intermediate state of RRM2 [15] linked with pathological misfolding. This intermediate state may enhance the access to the Nuclear Export Signal (NES) within its sequence, which increases the transport to the cytoplasm and serves as a molecular hazard linking physiological folding with pathological misfolding and aggregation.

A second effect of the RRM2 cleavage is the exposure to the solvent of its aggregation-prone *β*-strands [8,16]. These strands are normally buried in the native state, but have been found to form fibrils in vitro [9]. These processes confirm the role of the RRM2 fragment in the CTFs for the aggregation [16]. Since their possible causative role in the formation of the pathological condition is unclear, more investigations are needed. In particular, the *β*-strands could be at the core of the aggregation, because they are able to form steric zippers that, following a typical atomic model for amyloid fibril structure formation, give rise to amyloid structures [8]. Amyloid fibrils are formed by packed *β*-sheets that interact with each other through side chains. The side chains of neighboring sheets, projected roughly perpendicular to the fibril axis, interdigitate forming the so-called *steric zipper* [8]. In particular, it has been hypothesized that, specifically, the *β*3 and *β*5 strands form the steric zippers that originate the CTFs aggregation [8].

In support of the hypothesis that this kind of structure is at the base of the CTFs aggregation, as shown in Figure 1c1,c2, it has been recently found that some regions of RRM2 can form different classes of steric zipper structures [10,13] at the core of the formation of these amyloid fibrils [17].

Here, we use molecular dynamics (MD) simulations to explore the possible conformations of the ordered RRM2 regions of the CTFs. Leveraging on the MD simulation results, we suggest possible binding regions belonging to RRM2, which may be responsible for the fragment aggregations. The choice to analyze the conformational variation of each fragment independently from other possible partners is based on the hypothesis of the conformational selection model [18,19] validity. According to this theory, the bound conformations can be sampled by the protein even when it is not bounded to its corresponding partner: in other words, the conformational change of a protein can occur before a binding event, rather than being induced by the event itself [20].

This model also suggests that the right partner might act as a ‘molecular chaperon’ by stabilizing a non-pathological state: among the conformations of the dynamically fluctuating protein, this partner selects the one compatible with binding and shifts the conformational ensemble towards this state [21]. Moreover, the aggregation of TDP-43 is also influenced by the interaction with DNA and RNA [18]: indeed, RNA molecules can interfere with the aggregation kinetic, as a function of their nucleotides composition, binding affinity, and length [19].

To better understand the CTFs aggregation, here, we aim at finding the possible binding regions between the fragments.

To identify these binding regions, we use a method we have recently developed based on the mathematical formalism of the Zernike polynomial in two dimensions [22,23]. The method, characterized by a low computational cost compared to the commonly used version in three dimensions, examines portions of the molecular surface of two hypothetically interacting protein structures in terms of their local shape complementarity. Through an extensive sampling analysis of molecular patches, this superposition-free method is able to associate the probability of interaction between a patch and any other of the corresponding molecular partner. Using this formalism, we study the shape compatibility between different *β*-sheet regions as emerged from molecular dynamics frames. Moreover, to find the complexes binding poses, we perform docking selecting the poses whose binding regions met two requirements: a high number of involved *β*3 and *β*5 (two beta-sheets of the RRM2 domain) residues and high shape complementarity between the two surfaces, as calculated by Zernike descriptors. Finally, we perform additional 20 ns-long MD simulations to better characterize these interactions and analyze the molecular stability of the predicted complexes.

## 2. Results and Discussions

### 2.1. Molecular Dynamics Simulations and Equilibrium Conformations

To begin with, we carried out a standard MD simulation of 10 μs on both the considered fragments (see Methods section for details). In particular, the first fragment, Fragment A, corresponds to the residues 209–269 of TDP-43, while the second one, Fragment B, corresponds to the residues 220–269. Note that, for such small systems, the simulated time span could allow us to observe possible (perhaps intermediate) configurations that the fragments may explore after the cleavage in the cell, while this may not assure an exhaustive sampling of the configuration space at equilibrium of a typical protein.

Indeed, as we will discuss in the next section, we managed to observe a conformational transition passing through an unfolded state for Fragment B within the performed 10 μs simulation.

As a first analysis, we looked at the Root Mean Square Deviation (RMSD). Figure 2a1 shows that the RMSD of Fragment A has a steady behavior, with a mean value of 0.598 ± 0.065 nm. On the other hand, the RMSD of Fragment B, shown in Figure 2b1, is characterized by a sudden peak between 5.5 · 10^2^ ns and 6.5 · 10^2^ ns; this behavior is discussed in Section 2.2. To reduce the computational cost and facilitate the interpretation of the results, we selected the most representative structures in accordance with a clustering analysis. In particular, we firstly performed a Principal Component Analysis (PCA) of the covariance matrix of the atomic positions explored during the two MD simulations. Projecting each MD frame on the plane defined by the two first principal components, we obtained an essential representation of the fragment’s motions, shown in Figure 2a2 and Figure 2b2 for Fragment A and B respectively. Then, we executed a clustering analysis of the MD frames, according to such projection. Our aim is to find the most representative conformations, i.e., those closest to any other explored one, assumed by each simulated fragment at equilibrium. The appropriate number of clusters is evaluated by maximizing the Silhouette Coefficient (SC), a measure of cluster cohesion and separation (see Section 4.3 for more details).

Therefore, we selected the structures corresponding to the centroids of the clusters and studied such structures to search for possible binding regions with the Zernike polynomial formalism [22,23,24,25]. We found five equilibrium conformations (cluster centroids) for Fragment A (A1, A2, A3, A4, A5) and two for Fragment B (B1, B2). As shown in Figure 2a3,b3, conformations A2, A3, and A4 have similar secondary structures including a single *α* helix and three *β* strands. In particular, the *β* strands of A2 comprise residues 218–221, 229–234, and 254–256, for A3 218–221, 229–234, and 245–257, and for A4 218–221, 229–234, and 254–259. The *α* helix in all three conformations is at residues 237–244. Conformations A1 and A5 instead differ: the former with two *α* helices (at residues 237–244 and 267–260) and three *β* strands (at residues 218–221, 229–233, and 254–256), the latter with a single *α* helix (at residues 237–245) and four *β* strands (at residues 218–221, 229–234, 248–250, and 253–257). Conformation B1 has a *α* helix (237–245) and two *β* strands (229–231 and 256–258), whereas B2 has two *α* helices (237–246 and 253–258).

### 2.2. Unfolding Process of Fragment B

Looking at the time evolution of the RMSD in Figure 2b1, we can see that a peak is present around 6500 ns for Fragment B, which indicates the formation of an unstructured conformation. Indeed, we can observe in Figure 3 that, at the same time, the evolution of the total gyration radius *R_g_* also shows a peak. Since the *R_g_* of a protein is a measure of its compactness, this confirms the loss of the ordered structure of Fragment B and, therefore, the presence of a transition from the folded state to the unfolded state. While the fragment is stably folded (for about the first 5.5 μs and last 3.5 μs of the simulation), it maintains a relatively steady value of *R_g_*. On the other hand, in the interval between 5.5 and 6.5 μs, the *R_g_* abruptly increases, highlighting the unfolding process.

Interestingly, the fragment returns with rapid kinetics to a new equilibrium conformation. This result validates the advantages of working with small systems since the observation of fold-unfold transitions is computationally accessible. The transition causes a major conformational change, again characterized by a well-ordered structure, which could play a key role in the formation of the aggregation. Therefore, the structural rearrangement of fragment B, caused by protein cleavage, suggests investigating further possible fold-unfold transitions.

### 2.3. 2D Zernike Polynomial Expansion for Binding Regions Prediction

Recently, a new method based on the Zernike 2D polynomial expansion has been developed to evaluate whether and where two proteins can interact with each other to form a complex, based on their shape complementarity [22,23].

By expanding the solvent-exposed molecular surface patches in terms of 2D Zernike polynomials, it is possible to rapidly and quantitatively measure the geometrical complementarity between interacting proteins by comparing their molecular surfaces.

Here, we apply it to all the possible pairings between the 3D surfaces of the two CTFs RRM2 fragments, to find the binding regions on each surface. More specifically, for each point *i* belonging to the molecular surface, we define a molecular surface region (patch), which can be described through 2D Zernike formalism with a set of invariant descriptors. Two similar patches, if defined by the same reference point, have a small distance between the Zernike vectors, since perfectly complementary patches are equal under roto-translation. For each point *i* of one of the two surfaces, the distance between the Zernike descriptors of its patch and all the patches built on the points of the other surface is computed. The minimum of these values is selected, and, after all points had been studied, these minimum values are mapped in [0,1] and inverted. At the end of the process, points whose corresponding patches have high complementarity with the other surface are associated with a value of binding propensity (BP) near one. Thus, with a smoothing procedure, each point is associated with the mean BP value of the points in its neighborhood: the interacting regions should be made up mostly of elements with high complementarity and, therefore, a high average value of BP. Finally, we associate to each residue the mean BP value of the corresponding points in the surface. We apply this procedure to each surface in all the possible pairings.

At the end of this process, we obtain a set of BP profiles that we re-normalize computing the Z-score for each profile to identify more clearly the residues that are involved in the interaction.

Then, for each conformation in each pairing, we compute three means: the mean Z-score of the residues included in the β3 and β5 strands $m_{\beta 3,\beta 5}$, the mean Z-score of all the β-strands residues $m_{\beta }$, and the mean Z-score of the residues that are not part of β-strands $m_{r}$. This is because, while β-strands could, in general, give rise to amyloid structures, β3 and β5 are at the core of the interaction between CTFs according to our starting model [8]. The results are shown in Figure 4a, which reports the $m_{\beta 3,\beta 5}$, $m_{\beta }$, and $m_{r}$ values for each conformation in each pairing, or a wording when no β3 or β5 strands, or any kind of β-strand, are present. It can be observed that, except when A2 (which has not conserved the original β3 and β5 strands) and B2 (which has lost any β-strand) are considered, $m_{\beta 3,\beta 5}$ tends to have a higher value, whereas $m_{r}$ has always the lowest value. These results confirm our theory.

To describe the CTFs aggregation process, we are interested in finding the pairings in which both conformations have high $m_{\beta 3,\beta 5}$ values. With this aim, for each pairing, we compute the mean $\mu_{\beta 3,\beta 5}$ of the $m_{\beta 3,\beta 5}$ values of the two involved conformations. The result is depicted in Figure 4b. The pairings with the highest $\mu_{\beta 3,\beta 5}$ are the ones with a higher probability to be at the core of the CTFs aggregation according to the discussed model.

### 2.4. Molecular Docking for Complexes Binding Poses Prediction of Different Conformations

Since the procedure based on the Zernike method does not give us the complexes’ binding poses, we used the H-dock algorithm [26] on our best five pairings (with the highest $\mu_{\beta 3,\beta 5}$) to find possible bound conformations. For each pairing, we look at the first 20 docking poses of the two corresponding conformations (i.e., the ones obtained with our MD simulations) provided by the H-dock server. To select the docking predictions that present β3, β5 residues in the binding sites, we perform a contact analysis (see Methods for details). We then compute the percentage of β3, β5 residues involved in the bond and select the complexes associated with the highest values. The results are shown in Figure 5a. To associate these complexes to the pairings found with the Zernike algorithm, we search for the ones whose mean Z-score of the residues involved in the bonds ($m_{B}$) is higher than the mean Z-score of the residues not involved ($m_{nB}$), thus selecting the regions with the highest probability of interacting in accordance with the method based on the Zernike formalism (see Figure 5b). These results show that the binding sites identified with our method are not always predicted in the best complexes of the docking algorithm. Therefore, docking provides a set of possible molecular complexes but does not represent a replaceable solution to the approach used in this work, which is specifically based on the shape complementarity at the binding interface. However, by exploiting the optimization process of the binding pose performed by the docking method, we can propose structures derived by molecular docking in which the interacting regions are characterized by a high binding propensity predicted by the Zernike-based method.

### 2.5. Refining and Stability Analysis of the Selected Docked Complexes through MD Simulations

To characterize each proposed molecular complex, composed of two different fragment configurations, we performed statistical analysis on the dynamical behavior of each system, since it is known that good models have usually higher stability during MD simulations [27,28]. To this end, we performed 20 ns-long MD simulations [27] and analyzed the RMSD at equilibrium and the percentage of C*α* binding atoms in the docking complex that are preserved at the end of the simulation (as a function of increasing cut-off distance).

The results are shown in Figure 6, together with the interacting residues of the post-simulation complexes. These residues are defined as the ones including the C*α* atoms that are at a maximum distance of 8 Å from the other surface and are listed in Table 1. We note that, among the residues forming the binding region of the proposed complex A1–A3 (prediction 3), there is a target of post-translational modifications, specifically residue 263. In future work, it will be worth exploring the effects that those modifications can exert on both the fragments dynamics and their possible interactions, hopefully improving the understanding of the mutations’ role in the CTFs aggregation, which is still unclear [10]. Moreover, the effects of the RRM2 mutations, to the best of our knowledge, have not been studied specifically in the fragments constituting the CTFs, even if post-translational modifications can have distinct effects on different TDP-43 species [29].

Predictions 3 and 13 of the complex A1–A3 seem to be the most promising ones since they both preserve a high percentage of binding C*α* atoms and maintain a constant low value of the RMSD.

Indeed, MD simulations are a common tool to improve the quality of docking complexes by refining them [30,31,32], since they can account for conformational changes needed for binding at different levels, particularly on the scale of atoms, sidechains, loops, small molecules, or interfaces [27]. Thus, even complexes that lose part of initial (docking pose) contacts may assume a stable conformation during the dynamics refinement.

## 3. Conclusions

In this work, we aim at determining whether the portions of TDP-43 CTFs containing RRM2 fragments might be leading factors in their aggregation process.

Since the conformations that these fragments can assume have not yet been fully investigated, we began by studying the time evolution of the two possible RRM2 fragments constituting the CTFs, i.e., Fragment A and B, with MD simulations. Analyzing the trajectories of these two fragments, we found five representative conformations for Fragment A and two for Fragment B at equilibrium. Furthermore, we observed and characterized the unfolding of Fragment B.

Next, we searched on the surfaces of these equilibrium conformations possible regions of interaction, by verifying their shape complementarity, and associated each of these parings with a complex structure via a docking algorithm. We hypothesize that these complexes correspond to the structures most likely to be found in CTFs aggregates. Finally, to study the stability of these structures we performed additional MD simulations of 20 *ns*. We point out that almost all the proposed complexes have high stability compared to previous studies that used post-docking simulations to identify native conformations. Moreover, we note that even if several approaches to computationally predict a protein structure from its primary sequence have been developed, efficient exploration of the conformational space proteins can visit remains a hard task. In this respect, the reduced size of TDP43 fragments allowed us to exploit unbiased MD simulations to obtain conformations and to find a set of possible bound configurations, which may constitute the seeds for aggregation.

## 4. Materials and Methods

### 4.1. Dataset

The two starting structures for the MD simulations of Fragment A and Fragment B were extracted from the Protein Data Bank [33]. From the Nuclear Magnetic Resonance (NMR) structure of the TDP-43 tandem RRMs in complex with UG-rich RNA (PDB id: 4BS2) [34], we removed both the RNA and the RRM1 domain. The resulting structure, to which we refer as ‘whole RRM2’, contains residues from 192 to 269 of TDP-43. Next, to obtain the molecular structure of Fragment A, we removed residues up to the 208th, whereas, to obtain Fragment B, we removed the residues up to the 219th.

### 4.2. Molecular Dynamics Simulations

For both fragment structures, we carried out one molecular dynamics simulation of 10 μs. All steps of the simulation were performed using Gromacs 2019.3 [35].

The topologies of the system were built using the CHARMM-27 force field [36], the standard force field for proteins. Each fragment was placed in a rhombic dodecahedron simulative box, with periodic boundary conditions, filled with TIP3P water molecules [37]. The system of Fragment A and Fragment B included 4607 and 4668 water molecules, respectively. The rhombic dodecahedron box is built so that each atom of each fragment is at least at a distance of 11 Å from the box borders. This guarantees that approximately five layers of solvent molecules surround the fragment. The final system of Fragment A, consisting of 14,777 atoms, was minimized with 102 steps of steepest descent, whereas the system of Fragment B, consisting of 14,759 atoms, was minimized with 346 steps. Each step had a size of 0.01, while the force limit value was set to $max(|F_{n}\left|\right)<10^{3}$ kJ/mol/nm.

The thermalization and pressurization of the systems in NVT and NPT environments were run each for 0.1 ns at 2 fs time-step. The temperature was kept constant at 300 K with a Modified Berendsen thermostat and the final pressure was fixed at 1 bar with the Parrinello-Rahman algorithm [38]. A time constant of coupling between the system and the barostat of $\tau_{P}=2$ ps guarantees an average water density of 1006 ± 5 kg/m^3^ and 1002 ± 4 kg/m^3^, for Fragment A and B, respectively (close to the experimental value 1008 kg/m^3^). LINCS algorithm [39] was used to constraint h-bonds.

Finally, the systems were simulated with a 2 fs time-step for 10 μs in periodic boundary conditions, using a cut-off of 12 Å for the evaluation of short-range non-bonded interactions and the Particle Mesh Ewald method [40] for the long-range electrostatic interactions.

For all these steps, the Leap-Frog integrator and the Verlet cut-off scheme were used.

The same settings were used for the 20 *ns* simulations that were performed starting from the Zernike-selected docking complexes.

### 4.3. Principal Component Analysis and Clustering Analysis

To obtain an essential representation of the dynamics, we applied on the fragments’ trajectories a Principal Component Analysis (PCA) over the covariance matrix of the atomic positions [41]. To estimate the information preserved by projecting the trajectory on an essential *d*-dimensional space, we evaluated the Explained Variance Ratio (EVR) for each eigenvalue $\lambda_{i}$:(1)$$EVR\left(\lambda_{i}\right)=\frac{\lambda_{i}}{\sum_{j}^{3N}\lambda_{j}},$$
where *N* is the number of atoms. Since for both Fragment A and B, the first two eigenvalues result in a much higher EVR value compared to the other ones, we chose a two-dimensional projection (*d* = 2). To find the most representative conformations of the projection of each trajectory on its first two PCs, we implemented the k-means clustering algorithm. This algorithm has recently been employed on many MD simulations studies to reduce the dimensionality of the trajectories [42,43], by decreasing the number of structures while preserving essential structural/dynamical information. It results in the grouping of the MD conformations in similar structures, that are assumed to behave similarly. In particular, we selected the centroid of each cluster as a representative conformation for that class of structures. To evaluate the appropriate number of clusters (i.e., the value of *k*), the k-means clustering maximizes the Silhouette Coefficient (SC), which quantifies how well a data point fits into its assigned cluster. For each point *i* in a cluster $C_{i}$, it defines a silhouette value:(2)$$s\left(i\right)=\left\{\begin{array}{cc} \frac{b(i)-a(i)}{max\left(a(i),b(i)\right)}, & \mathrm{if}\;|C_{i}|\;>1\\ 0, & \mathrm{if}\;|C_{i}|\;=1, \end{array}\right.$$
where a(i) is called similarity and is defined as
(3)$$a\left(i\right)=\frac{1}{|C_{i}|-1}\sum_{\underset{i\neq j}{j\in C_{i}}}d\left(i,j\right),$$
with d(i,j) the distance between data points $x_{i}$ and $x_{j}$ in the cluster $C_{i}$. b(i) is the dissimilarity and is defined as
(4)$$b\left(i\right)=\underset{k\neq i}{min}\sum_{j\in C_{k}}d\left(i,j\right).$$*s(i)* ranges between −1 and 1; a value near one indicates that the point has been clustered appropriately. The mean s(i) over all data of the entire dataset, $\tilde{s}$, is a measure of how appropriately the data have been clustered. The maximum value of the mean over all data of the entire dataset is the SC.

Table 2 shows the mean silhouette value $\tilde{s}$ for different *k* number of clusters, for the two fragments.

### 4.4. Computation of Molecular Surfaces

The molecular surfaces were obtained starting from the PDB files found after the clustering of the PCA of the trajectories resulting from the MD simulations. To compute the solvent-accessible surface for the considered structures, we used DMS [44], with a density of 5 points per Å${}^{2}$ and a water probe radius of 1.4 Å. For each surface point, we calculated the unit normal vector with the flag −n.

### 4.5. Evaluation of Shape Complementarity

The first step of this algorithm is to select from the surface a patch Σ, defined as the set of surface points that fall within a sphere of radius $R_{zernike}=6$ Å centered on one point of the surface. The points contained in this sphere are divided, with a clustering from a random point that includes only the points closer than a distance $D_{p}$, in points belonging to the surface and points not directly connected to it (for example coming from a protuberance included in the sphere). Only the former will constitute the patch. Once the patch has been selected, the mean vector of the normal vectors of the patch points is computed and oriented along the *z*-axis. Thus, given a point *C* on the *z*-axis, we define θ as the largest angle between the *z*-axis and a secant connecting *C* to any point of the patch Σ. *C* is then set so that $\theta =45^{\circ }$, and each surface point is labeled with its distance *r* from *C*. As a next step, a square grid is built, where each pixel is associated with the mean of the *r* values associated with the points inside each pixel. The resulting 2D function is then expanded on the basis of the Zernike polynomials. Indeed, each function of two variables f(r,ψ) defined in polar coordinates inside the region of the unitary circle can be decomposed in the Zernike basis as
(5)$$f\left(r,\psi \right)=\sum_{n^{\prime }=0}^{\infty }\sum_{m=0}^{n^{\prime }}c_{n^{\prime }m}Z_{n^{\prime }m}\left(r,\psi \right),$$
with
(6)$$c_{n^{\prime }m}=\frac{n^{\prime }+1}{\pi }\int_{0}^{1}dr\;r\int_{0}^{2\pi }d\psi Z_{n^{\prime }m}^{*}\left(r,\psi \right)f\left(r,\psi \right),$$
and
(7)$$Z_{n^{\prime }m}=R_{n^{\prime }m}\left(r\right)e^{im\psi }.$$$c_{n^{\prime }m}$ are the expansion coefficients, while the complex functions $Z_{n^{\prime }m}\left(r,\psi \right)$ are the Zernike polynomials. The radial part $R_{n^{\prime }m}$ is given by
(8)$$R_{n^{\prime }m}\left(r\right)=\sum_{k=0}^{\frac{n^{\prime }-m}{2}}\frac{{\left(-1\right)}^{k}\left(n^{\prime }-k\right)!}{k!\left(\frac{n^{\prime }+m}{2}-k\right)!\left(\frac{n^{\prime }-m}{2}-k\right)!}.$$

Since, for each couple of polynomials, it is true that
(9)$$<Z_{nm}|Z_{n^{\prime }m^{\prime }}>=\int_{0}^{1}dr\;r\int_{0}^{2\pi }d\psi Z_{nm}^{*}\left(r,\psi \right)Z_{n^{\prime }m^{\prime }}\left(r,\psi \right)=\frac{\pi }{n+1}\delta_{nn^{\prime }}\delta_{mm^{\prime }},$$
the complete sets of polynomials form a basis, and knowing the set of complex coefficients $c_{n^{\prime }m}$ allows for a univocal reconstruction of the original patch. Once a patch is represented in terms of its Zernike descriptors, the similarity between that patch and another one can be simply measured as the Euclidean distance between the invariant vectors. The norm of each coefficient $z_{n^{\prime }m}=\left|c_{n^{\prime }m}\right|$ constitutes one of the Zernike invariant descriptors. Since $z_{n^{\prime }m}$ does not depend on the phase (i.e., it is invariant for rotations around the origin of the unitary circle), two patches can be assessed by comparing the Zernike invariants of their associated 2D projections, without considering their orientation. On the other hand, the relative orientation must be taken into account: if we search for similar regions we must compare patches that have the same orientation once projected in the 2D plane, i.e., the solvent-exposed part of the surface must be oriented in the same direction for both patches (for example as the positive z-axis). If instead, we want to assess the complementarity between them, we must orient the patches contrariwise, i.e., one patch with the solvent-exposed part toward the positive z-axis (‘up’) and the other toward the negative *z*-axis (‘down’).

Thus, to assess whether two surfaces have regions with a relevant shape complementary, we (i) compute the Zernike descriptors of the patches centered in all the points of the two surfaces up to the selected maximum expansion order *n* (The two surfaces have to be oriented in opposite verse along the *z*-axis.). (ii) For each point *i* of the two surfaces, we compute the distance between the Zernike descriptors of the patches of one surface and all the patches built on the points of the other surface. The minima of these values are selected. Next, the found minima values are normalized in the range [0,1] and inverted so that higher values correspond to higher shape complementarity matches [22]. (iii) Finally, we perform a smoothing process, where each point is associated with a final binding propensity (BP) computed as the mean value of the points in its neighborhood, defined as all the points having a spatial distance from it smaller than 6 Å.

### 4.6. Contact Analysis

To recognize the interaction interface residues in the H-dock predicted complexes, we looked at the position of Cα atoms. For each structure in the complex, we selected the Cα atoms that are at a distance smaller than 9 Å from the Cα atoms of the other structure [45,46]. To determine the interacting patches, we select the residues corresponding to the atoms included in a sphere of radius 3 Å centered on these Cα atoms.

## Figures and Tables

**Figure 1 biomolecules-11-01905-f001:**
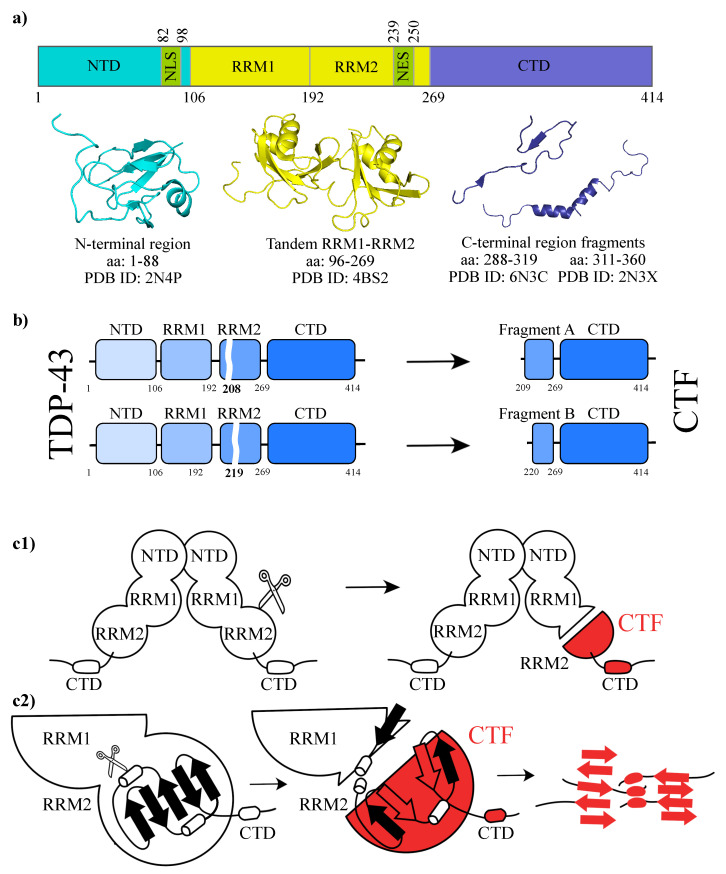
Structural organization of TDP-43 and hypothesized model for CTFs aggregation. (**a**) TDP-43 comprises an NTD, two RRM domains, a nuclear export signal (NES), a nuclear localization signal (NLS), and a disordered C-terminal domain. The cartoon representation of the structure of the NTD portion comprising residues 1–88 (PDB ID: 2N4P) is shown in blue, the structure of residues 96–269 of the tandem RRM1-RRM2 domains (PDB ID: 4BS2) in yellow. Two fragments of the CTD are reported in violet, counting residues 288–319 (PDB ID: 6N3C) and 311–360 (PDB ID: 2N3X). (**b**) Schematic representation of the two possible cleavages of TDP-43 (at sites 208 or 219) from which the CTFs can be originated. The two truncated RRM2 fragments are called Fragment A and B, respectively. (**c1**) After the cleavage, the CTF is split from the whole TDP-43, which, in physiological conditions, forms dimers. (**c2**) Scheme of the hypothesized aggregation model. The RRM2 fragment resulting from the cleavage exposes its β-strands; the β-strands from different CTFs allow the formation of aggregates to happen.

**Figure 2 biomolecules-11-01905-f002:**
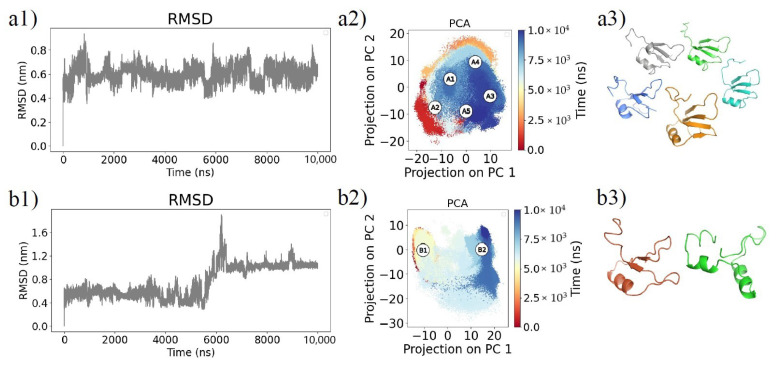
Analysis of the Molecular Dynamics simulations. (**a1**) Root Mean Square Deviation (RMSD) as a function of time for Fragment A structure with respect to its starting state. (**a2**) Two-dimensional projection of the sampled conformations in the subspace spanned by the first two Principal Components (PCs) of the covariances of the atomic positions during the simulation. Points are colored from red to blue as simulation time goes from zero to 10 μs. White circles mark the most representative conformations according to cluster analysis (see Section 4.3). (**a3**) Cartoon representation of representative conformations marked in (**a2**). (**b1**–**b3**): same as in (**a1**–**a3**), respectively, but for Fragment B.

**Figure 3 biomolecules-11-01905-f003:**
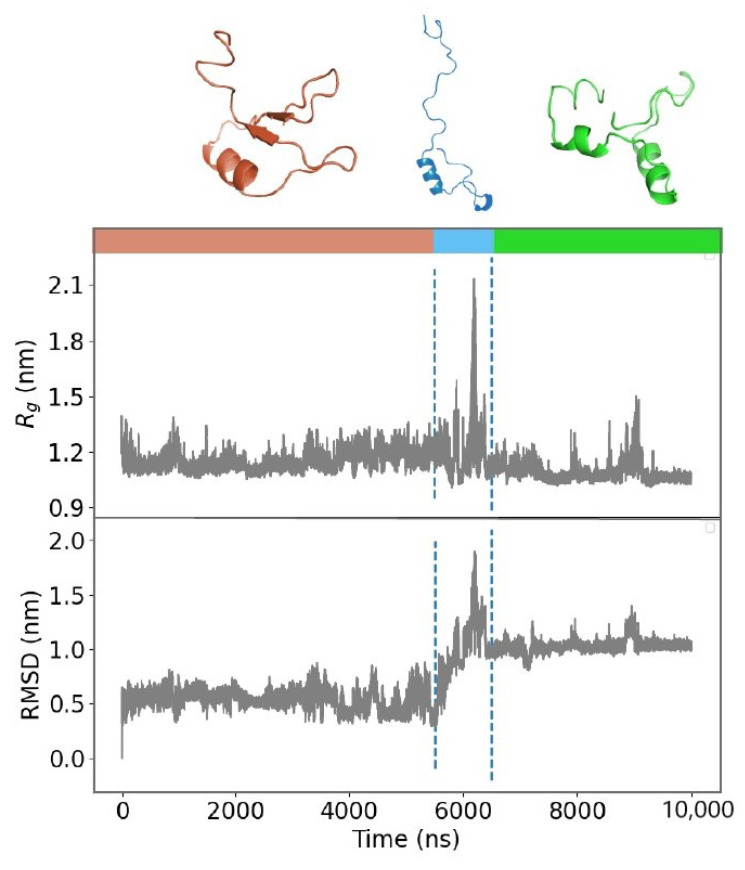
Analysis of the unfolding process of Fragment B. Time evolution of the Radius of Gyration ($R_{g}$) and Root Mean Square Deviation (RMSD) for Fragment B. The time span is divided into three parts (marked by the dotted lines at approximately 5.5 and 6.5 μs), according to the different overall organization displayed by the fragment during the dynamics. Representative cartoon representations of the fragment structure are reported on the upper side of the figure.

**Figure 4 biomolecules-11-01905-f004:**
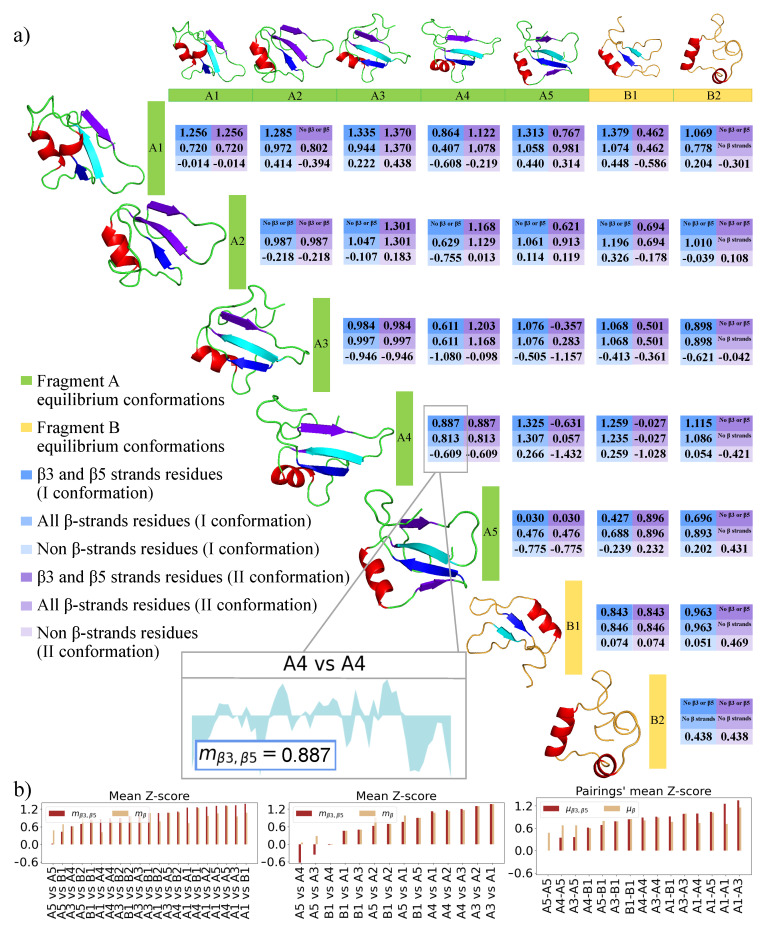
Analysis of the shape complementarity between the sampled MD conformations. (**a**) Comparison between all possible pairings of the five extracted conformations for Fragment A (green labels) and the two extracted conformations for Fragment B (yellow labels). For all possible pairings between conformations, we compute the BP of the residues of both surfaces and report the mean of the Binding Propensity (PB) Z-score of the residues that are part of the β3 and β5 strands ($m_{\beta 3,\beta 5}$, in dark blue for the first surface in the pairing and in dark purple for the second one), of all the β-strands ($m_{\beta }$, in blue for the first surface and in purple for the second) and of the residues that are not part of β-strands ($m_{r}$, in light blue for the first surface and light purple for the second). An example of a Z-score profile for all the residues of conformation A4 compared with itself is shown in the zoom. Cartoon representations of the Fragments’ conformations are shown in correspondence with the reported scores. For each conformation, the β3 and β5 residues are colored in cyan or blue, respectively. The remaining β residues are marked in purple, while the α residues are colored in red. (**b**) From left to right, bar plots representation of the $m_{\beta 3,\beta 5}$ and $m_{\beta }$ values computed for the first conformation in each pairing of (**a**), for the second, and means ($\mu_{\beta 3,\beta 5}$, $\mu_{\beta }$), over the two conformations. The pairings are ordered from left to right according to increasing $\mu_{\beta 3,\beta 5}$ values.

**Figure 5 biomolecules-11-01905-f005:**
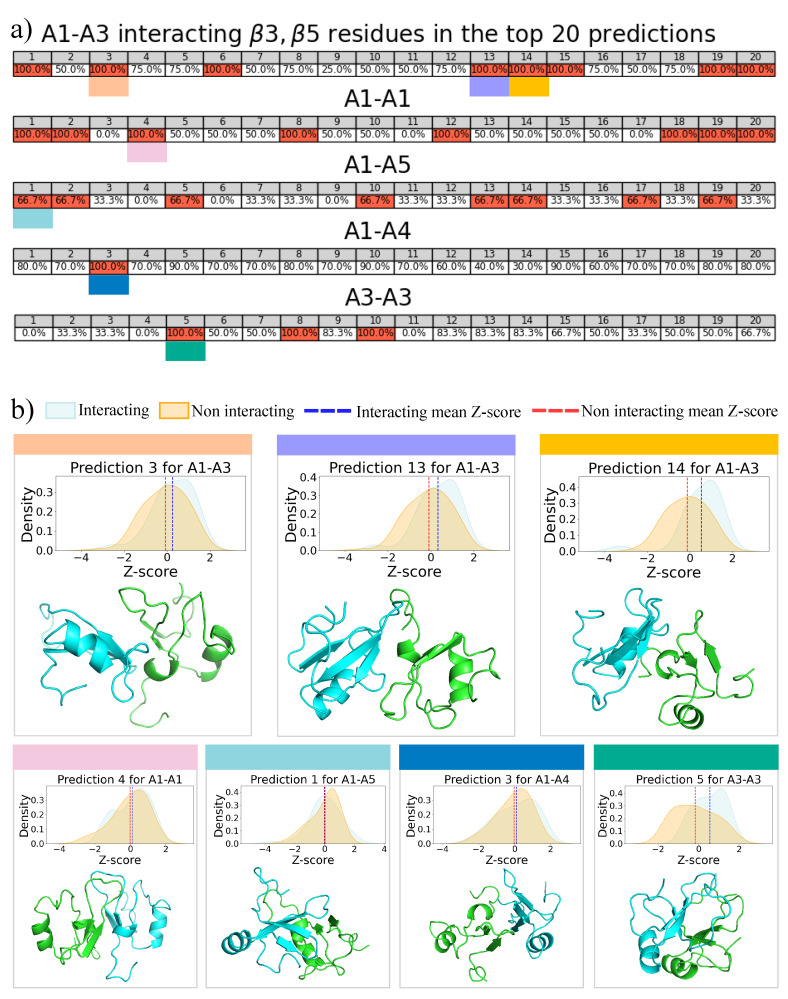
Analysis and selection of Fragments’ binding poses. (**a**) Percentages of β3, β5 residues involved in the bond over the total number of β3, β5 residues for the top 20 docking-predicted complexes comprising the top five Zernike-selected pairings. Predictions with the highest percentages are underlined in red, while colored boxes point out the complexes characterized by a mean Binding Propensity (BP) Z-score of the residues involved in the bond ($m_{B}$) higher than the one computed for the non-interacting residues ($m_{nB}$). (**b**) Probability distribution of BP Z-scores for interacting (cyan) and non-interacting (orange) residues for each prediction marked by a colored box in panel (**a**). Vertical blue and red line represent the $m_{B}$ and $m_{nB}$ values, respectively. Cartoon representation of the analyzed complexes are reported below the BP Z-score distributions.

**Figure 6 biomolecules-11-01905-f006:**
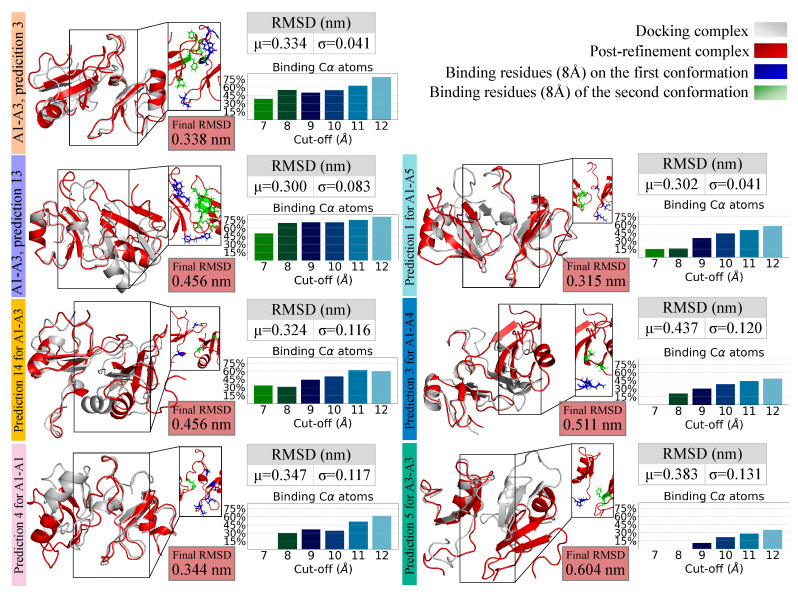
Comparison between docking complexes before and after the MD simulation. Cartoon representation of each of the seven Zernike-selected docking complexes (grey) and of the corresponding structure obtained after a MD simulation of 20 ns (red). Zooms display the binding residues on the first (blue sticks) and second (green sticks) conformation after the MD simulation. For each complex, the mean (μ) and standard deviation (*σ*) of the RMSD during the simulation (excluding the first two ns) are reported. Light-red boxes show, instead, the RMSD value of the structure at the end of the simulation. A bar-plot with the percentage of the C*α* binding atoms in the docking complex that are preserved at the end of the simulation (as a function of increasing cut-off distance) is reported as well.

**Table 1 biomolecules-11-01905-t001:** List of the residues found in contact in the post-MD simulation complexes. Interacting residues are defined as the one whose C*α* atoms have a distance lower than 8 Å with the partner C*α* atoms.

**A1–A3,**	I conformation	223, 259, 260, 261, 262, 263
**prediction 3**	II conformation	225, 226, 227, 228, 229, 256
**A1–A3,**	I conformation	224, 225, 259, 260, 261, 262
**prediction 13**	II conformation	227, 228, 229, 247, 248, 249, 252, 253, 254, 267, 268, 269
**A1–A3,**	I conformation	227, 259
**prediction 14**	II conformation	229, 246
**A1–A1,**	I conformation	221, 223, 231, 260
**prediction 4**	II conformation	258, 259
**A3–A3,**	I conformation	260, 261
**prediction 5**	II conformation	227, 228, 229
**A1–A5,**	I conformation	221, 223, 259
**prediction 1**	II conformation	221, 222, 225, 226
**A1–A4,**	I conformation	223, 224
**prediction 3**	II conformation	247, 254

**Table 2 biomolecules-11-01905-t002:** Mean Silhouette Coefficient, $\tilde{s}$, for different number of clusters, *k*, for Fragment A and B.

Number of Clusters	Fragment A	Fragment B
*k* = 2	0.4489	0.7095
*k* = 3	0.4568	0.7066
*k* = 4	0.5023	0.6795
*k* = 5	0.5148	0.6393
*k* = 6	0.4833	0.6735

## Data Availability

Not applicable.

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
