# Peer review of "A Computational Approach to Investigate TDP-43 RNA-Recognition Motif 2 C-Terminal Fragments Aggregation in Amyotrophic Lateral Sclerosis"

_biomolecules, 2021, doi:10.3390/biom11121905_

Round 1
Reviewer 1 Report
- Both simulations are clearly very far from convergence. Converging such systems might be very hard (or even impossibile in a short time frame) but this should be clearly stated in the text.
- The PC projections show that there are no clusters (hence no metastable states) observed during simulations. This makes the cluster analysis not very meaningful. Also, it is not clear how the number of clusters was selected.
- The authors should at least describe the differences between centroids e.g. in terms of secondary structure
- By generating fragments individually, the authors assume a conformational selection type of scenario, where no remodelling occurs after binding. This might be not necessarily true: perhaps it could be useful to discuss this point either in the introduction or in the conclusions (or both).
- I would suggest the authors to improve the readability of fig 4, which is central in this work. Showing the regions b3, b5, etc, onto the 3d structures could be very useful. What is the scale/unit of the binding propensity?
- The authors suggest A1-A3, A1-A1, A3-A3, A1-A5 and A1-A4 as potential complexes. These structures are not shown, nor described in the text, making it impossible for the reader to understand these interactions. While I can understand that the viability of these complexes might be difficult to prove experimentally, the authors could run MD simulations from docked poses to check if these are at least stable computationally.
Author Response
please find our comments in attached file.

Reviewer 2 Report
The present manuscript “A computational approach to investigate TDP-43 RNA-recognition motif 2 C-terminal fragments aggregation in Amyotrophic Lateral Sclerosis” submitted by Greta Grassmann and coworkers reports a bioinformatics study on TDP-43 RNA recognition motif 2. TDP-43 is a biomedically relevant protein, which has been linked to the formation of insoluble cytoplasmic aggregates and development of Amyotrophic Lateral Sclerosis (ALS). In this work, the authors have applied extensive Molecular Dynamics simulations to investigate the aggregation process of TDP-43 C-terminal fragments. Lastly, they also studied the protein-protein interactions by using molecular docking analyses. Overall, the manuscript is well-presented, and the topic addressed is interesting for the field. However, in my opinion, the manuscript is preliminary and needs further improvements before publication in Biomolecules.
Major and minor comments
- This manuscript reports a bioinformatics study without in vitro/in vivo evidence that demonstrates the proposed mechanisms. In my opinion, the quality of the manuscript would improve if the authors can add experimental data to support the proposed mechanism of aggregation and mode of interaction.
- In this manuscript the authors used an interesting approach for molecular dynamics calculations based on based on Zernike polynomials. They modeled the possible aggregation of two different C-terminal fragments (A and B) of TDP-43. These fragments corresponds to the segments 209-269 or 220-269 of TDP-43, for A and B, respectively. They termed such fragments as CTFs. This approach is valid, however the C-terminal disordered region of TDP-43 is not included in their calculations. How did the authors consider a possible effect of this intrinsically disordered region on the CTFs aggregation? Do they think that folding-upon-binding mechanisms can happpen in this case? Please discuss this in the manuscript.
- Please explain how post-translational modifications of TDP-43 can affect CTF aggregation based on your models.
- Line 39. Please edit the text to make clear that “195-206“ indicates the lenght in residuesof the C-terminal fragments.
- Related to my previous comment, the incorporation of a structural model of TDP-43 might help the readers to visualize the location of the C-terminal fragments analysed under a conformational context.
- Lines 219-221. The authors mention that “ the structure of these fragments has not yet been fully investigated and their conformations are not yet available“. Please specify which conformations are not available. Recently, current AI tools have been developed for structure prediction as well as for evaluating multimer predictions. Please discuss the adavantages of your method compared with these recently-developed approaches.
Author Response

(The authors gave the same response as above.)

Round 2
Reviewer 1 Report
We thank the authors for the efforts in improving the manuscript.
This reviewer still has the following concerns:
- As a general comment, the new text contains several typos and the manuscript could benefit from proofreading.
- I still find the cluster analysis a bit odd. The PCA shows that samples do not cluster into groups (at least using the first two PC projection). Hence, any attempt to cluster the data will not be very meaningful. In essence, my suspicion is that the cluster analysis may not be necessary at all in this case. At this stage my suggestion is to remove the coloured scatter plots (fig 2.a.iii and 2.b.iii ) and place the centroid labels on the PCA projections (2.a.ii and 2.b.ii ) or on the RMSD plots. Note that the unit is missing on the color bars and PC should be unitless. Please check that all labels in the figure and caption are present and described.
P2 .Line 31 “Alzheimer’s Disease (AD) [2,3], even if, however, “ please revise
P2,In the following study -> In this study
P2. cysteine residue 244 and the lysine residue 244. Please revise
P3. Caption: “The PDB structures of a fragment of the NTD (PDB ID: 2N4P), of the tandem RRM1-RRM2 (PDB ID: 4BS2) and of two fragments of the CTD (PDB ID: 6N3C and 2N3X are reported as well.” This sentence is not clear. Please revise.
P3. Caption. The labeling does not correspond to the text in the caption
P4. L90: resuts -> results. L92. conformation -> conformational. The entire sentence is not clear.
P5. L136: exhaustively -> exhaustive. L 143: unfold -> unfolded
P11. "The gray box shows instead the RMSD value of the structure at the end of the simulation" I can’t seem to find the final RMSD in the figure
P12. L282: MD are -> MD simulations are (?)
Reviewer 2 Report
The manuscript can be accepted in present form. All the drawbacks present in the first version of the manuscript have been carefully adressed by the authors by either providing new experiments and relevant discussions.
Author Response
We thank the Reviewer for appreciating the work we did in response to the proposed comments/suggestions.